# Ten-Year Trends in Hepatocellular Carcinoma Mortality: Examining the Interaction Between Fibrosis Score and Patient Age

**DOI:** 10.3390/diseases13080256

**Published:** 2025-08-12

**Authors:** Ayrton Bangolo, Hadrian Hoang-Vu Tran, Budoor Alqinai, Rishabh Goyal, Shehwar Ahmed, Aamna Qasim, Gabriela Rojas, Shubham Madan, Helena Barbosa, Zainab Mustafa, Risham Waseem, Gabriel Ingersoll, Hamza Khan, Alison Guzzetti, Jonathan Daniel, Samiya Parkar, Aakriti Tiwari, Sarah Lafleur, Rajasekhar Cingapagu, Saliha Y. Amasyali, Eric Pin-Shiuan Chen, Simcha Weissman

**Affiliations:** 1Department of Hematology and Oncology, John Theurer Cancer Center, Hackensack, NJ 07601, USA; 2Department of Internal Medicine, Hackensack Palisades Medical Center, North Bergen, NJ 07047, USA

**Keywords:** hepatocellular carcinoma, liver fibrosis, cirrhosis, age interaction, SEER database, cancer-specific mortality, overall mortality, prognostic factors, survival analysis, multivariate Cox regression

## Abstract

Background: Hepatocellular carcinoma (HCC) remains a major cause of cancer-related mortality worldwide, with survival outcomes influenced by a range of demographic and pathological factors. While cirrhosis is a well-established risk factor, recent evidence shows that HCC can also develop in patients with only mild to moderate liver fibrosis. However, there is limited understanding of how fibrosis severity interacts with other clinical variables, such as patient age, to affect mortality. This study aims to explore how fibrosis scores relate to both overall and cancer-specific mortality in US HCC patients, with an emphasis on how this relationship may shift across different age groups. Methods: We utilized data from the Surveillance, Epidemiology, and End Results (SEER) database to identify 15,796 adult patients diagnosed with HCC between 2010 and 2021. Baseline demographics, disease characteristics, and treatment variables were examined. Mortality outcomes were evaluated using Cox proportional hazard regression. Variables significant at *p* < 0.1 in univariate analysis were included in multivariate models to identify independent predictors of mortality (with hazard ratios [HRs] > 1 signifying increased risk). A secondary analysis assessed how age modifies the association between fibrosis score and mortality. Results: The study population was predominantly male (77.2%), with most patients aged 60–79 (59.6%) and presenting with localized disease (61%). A majority had advanced liver fibrosis or cirrhosis (81.7%) and lived in large urban areas (62.9%). Crude comparisons indicated that male sex, older age, single status, advanced tumor stage, lower income, and cirrhosis were linked to worse outcomes. In adjusted models, independent predictors of increased mortality included male sex, older age, unmarried status, and more advanced disease stage. Receipt of surgery or chemotherapy was associated with a lower risk of death. Notably, the influence of fibrosis on mortality was found to be greater in older patients than in their younger counterparts. Conclusions: This analysis identifies key prognostic indicators in HCC and suggests that the relationship between fibrosis and survival is not uniform across age groups. These findings support the need for age-specific clinical management strategies and highlight the potential benefit of early detection and appropriate interventions, even in non-cirrhotic patients.

## 1. Introduction

HCC is a significant global health concern and remains one of the leading causes of cancer-related deaths worldwide. HCC accounts for approximately 80% of primary liver cancers and is often diagnosed at an advanced stage, which typically results in poor prognosis and high mortality rates [1]. However, it is important to note that the stage at diagnosis can vary widely depending on the patient population and clinical setting. For example, in population-based registries such as the Surveillance, Epidemiology, and End Results (SEER) program, a substantial proportion of patients may be diagnosed at an earlier stage and thus be eligible for surgical intervention. In our study cohort of 15,796 patients, 94.4% underwent surgery, reflecting that many patients had localized or early-stage disease amenable to curative treatment. This highlights a difference from some prior reports and underscores the heterogeneity of HCC presentation across populations.

Traditionally, HCC has been closely linked to cirrhosis, a condition characterized by severe liver fibrosis resulting from chronic liver diseases such as hepatitis B and C and excessive alcohol consumption [2,3,4,5]. However, recent studies have reported an increasing incidence of HCC among patients with mild to moderate fibrosis, indicating a broader spectrum of liver disease severity contributing to HCC development [6,7].

Despite the well-documented association between HCC and cirrhosis, there remains limited understanding of how the degree of fibrosis influences HCC mortality, particularly in the context of other independent prognostic factors such as age, gender, and tumor stage. Previous research has predominantly focused on overall survival rates without thoroughly examining how varying fibrosis levels affect mortality outcomes [8,9,10,11]. Understanding the complex interplay between fibrosis severity and other demographic and clinicopathologic characteristics is crucial for developing tailored management strategies aimed at improving patient outcomes.

In this study, we aimed to explore how varying levels of liver fibrosis influence mortality outcomes among individuals diagnosed with HCC in the United States over a recent 12-year period. Special attention was given to how age may modify the impact of fibrosis on survival. Using data from the SEER program, we conducted an in-depth evaluation of patient demographics, tumor characteristics, and treatment modalities to identify determinants of both overall and HCC-specific mortality. Through multivariate Cox proportional hazard regression analyses, we aimed to delineate the independent and interactive effects of fibrosis score and age on HCC mortality, thereby providing insights that could inform clinical decision-making and personalized patient care.

## 2. Methods

### 2.1. Study Design

We carried out a retrospective cohort study using data from the Surveillance, Epidemiology, and End Results (SEER) Program. This population-based database collects cancer incidence, demographic, clinicopathological, and survival information from 17 US cancer registries and encompasses approximately 28% of the national population. Data were extracted from the November 2023 submission and accessed via the website (http://www.seer.cancer.gov) accessed on 20 May 2024.

### 2.2. Cohort Selection

#### 2.2.1. Inclusion Criteria

The study included individuals diagnosed with hepatocellular carcinoma (HCC) between 1 January 2010 and 31 December 2021. Selection was based on specific primary tumor locations and histologic codes corresponding to HCC.

#### 2.2.2. Exclusion Criteria

Cases were excluded if key variables were missing, including age at diagnosis, race, fibrosis score, or cancer staging information.

### 2.3. Variables of Interest

#### 2.3.1. Primary Exposures

All collected variables were treated as exposures, with a primary interest in examining how the interaction between age and hepatic fibrosis severity might influence outcomes.

#### 2.3.2. Outcome Measures

Overall Mortality (OM): Defined as death from any cause during the study period. Patients were categorized as “deceased” or “alive” based on their vital status at the last follow-up.

Cancer-Specific Mortality (CSM): Deaths specifically attributed to HCC were classified as “yes,” while deaths from unrelated causes were categorized as “no.” Yearly cancer-specific death counts from 2010 to 2021 were analyzed.

#### 2.3.3. Survival Time

For overall survival analysis, duration was calculated from the date of initial diagnosis to the date of death or last recorded follow-up (censored on 31 December 2023). For CSM, survival was measured up to the date of HCC-related death or last follow-up if the cause of death was not HCC.

#### 2.3.4. Demographic and Clinical Characteristics

The study extracted the following variables: age at diagnosis, sex, racial/ethnic background (categorized as White, Black, or Other—which includes American Indian/Alaska Native and Asian/Pacific Islander), Hispanic origin, year of diagnosis, stage at diagnosis (localized, regional, distant), fibrosis score, geographic residence type, annual household income, marital status, and treatment modalities (surgery, radiation, chemotherapy). The fibrosis score was derived from the SEER variable ‘Ishak fibrosis score’, where available, categorized as 0–4 (no cirrhosis) and 5–6 (cirrhosis). This binary categorization is based on prior validation studies that dichotomize early vs. advanced fibrosis using histologic staging.

#### 2.3.5. Treatment Details and Limitations

While 94.4% of patients in our cohort underwent surgery and 61.02% were diagnosed at the localized stage, it is important to note that patients with a non-localized disease (regional or distant stage) may also undergo various surgical procedures, including palliative resections or liver transplantation. However, the SEER database does not provide detailed information on the specific types or intent of surgeries performed. SEER records surgical treatment as a general category and does not distinguish between specific procedures such as partial hepatectomy or liver transplantation. Therefore, we were unable to evaluate differences in outcomes based on surgical type. This limitation restricts the interpretation of surgical effects, particularly given the prognostic and therapeutic differences between resection and transplantation. Similarly, chemotherapy data in SEER lack granularity regarding treatment type and regimen, preventing a distinction between conventional cytotoxic agents, molecular targeted therapies, or immunotherapies. This limitation constrains the interpretation of treatment effects, especially in an era when targeted and immune-based treatments have become standard for advanced HCC.

Additionally, some patients may receive combined treatments involving surgery, chemotherapy, and/or radiotherapy. Due to these data limitations, our analysis cannot distinguish curative from palliative surgeries nor fully characterize multimodal treatment patterns. This limitation is acknowledged and should be considered when interpreting treatment-related findings. Treatment modalities, including surgery, chemotherapy, and radiotherapy, were recorded as separate variables. Due to the nature of the SEER database, patients may have received one or more of these treatments either sequentially or concurrently. We analyzed the frequency of patients receiving single, dual, or triple combinations of these therapies where possible. However, detailed timing and intent of combined treatments were not available.

### 2.4. Statistical Methods

We applied Cox proportional hazard regression modeling, under the assumption of time-invariant hazard ratios, to identify factors associated with OM and CSM. Variables showing a univariate association with a *p*-value < 0.1 were entered into a multivariable model (Model 1). A second model (Model 2) incorporated the interaction term between age and fibrosis score to further assess their combined prognostic impact. Hazard ratios (HRs) greater than 1 were interpreted as indicators of elevated risk. Two-sided tests were used throughout, with significance set at *p* < 0.05 and 95% confidence intervals. All analyses were conducted using STATA version 18.

## 3. Results

Table 1 presents the demographic and clinicopathologic characteristics of 15,796 patients diagnosed with hepatocellular carcinoma (HCC) in the United States from 2010 to 2021. Males accounted for 77.23% of cases. The majority of patients were diagnosed between the ages of 60 and 79 (59.62%), followed by those aged 40–59 (33.90%). Approximately half of the patients (50.31%) were married at diagnosis, with the remainder being single (26.73%), divorced/separated (15.35%), or widowed (7.61%). Most tumors were diagnosed at a localized stage (61.02%), with 28.94% regional and 10.04% distant stage. Non-Hispanic Whites constituted the largest racial group (47.78%), followed by Hispanics (23.31%), Non-Hispanic Blacks (10.99%), and other races (17.92%). The majority of patients (62.95%) lived in metropolitan counties with populations over 1 million, with the remainder residing in smaller metropolitan or nonmetropolitan areas. Regarding income, 35.76% of patients earned $60,000–79,999, and 30.53% earned $80,000–99,999 annually. Most patients (81.74%) had cirrhosis. Treatment modalities included surgery in 94.44% of patients, chemotherapy in 44.25%, and radiation therapy in 12.42%. The annual number of diagnoses fluctuated over the study period, with a marked decline in 2020 and 2021, potentially reflecting external influences such as the COVID-19 pandemic.

Table 2 outlines key findings from the unadjusted analysis of variables linked to overall and cancer-specific mortality in US patients diagnosed with HCC between 2010 and 2021. Males had higher mortality risks than females, with hazard ratios (HRs) of 1.19 for overall and 1.20 for cancer-related mortality. Older age at diagnosis was associated with increased mortality, particularly in patients aged 80+ (HRs of 3.08 for overall and 2.78 for cancer-related mortality). Unmarried patients, especially widowed individuals, had higher mortality risks. Advanced tumor stages greatly increased mortality risks, with distant stage having HRs of 4.92 for overall and 5.74 for cancer-related mortality. Non-Hispanic Black patients had a slightly higher risk of overall mortality, while those categorized as “Other” had lower mortality risks. Living in nonmetropolitan areas generally correlated with higher mortality. Higher income was associated with lower mortality, particularly for those earning $100,000+ (HRs of 0.70 for both overall and cancer-related mortality). Patients with cirrhosis, those who received radiation, and those who underwent chemotherapy had higher mortality risks, whereas surgery significantly reduced mortality (HRs of 0.48 for overall and 0.47 for cancer-related mortality).

The multivariate Cox proportional hazard regression analysis (Table 3) of US patients diagnosed with hepatocellular carcinoma (HCC) between 2010 and 2021 reveals several significant factors affecting all-cause and cancer-related mortality. Males had higher adjusted proportional hazard ratios (AHRs) for both overall mortality (1.18) and cancer-related mortality (1.17) compared to females. Older age at diagnosis was associated with increased mortality, particularly for those aged 80+ (AHRs of 3.25 for overall and 2.95 for cancer-related mortality). Unmarried patients, including single, divorced/separated, and widowed individuals, showed higher mortality risks than married patients. Advanced tumor stages significantly elevated mortality risks, with distant stage having AHRs of 4.81 for overall and 5.57 for cancer-related mortality. Racial analysis showed that patients classified as “Other” had a slightly lower mortality risk. Living in less populous metropolitan areas or nonmetropolitan areas adjacent to metropolitan areas was associated with higher cancer-related mortality. Income levels did not show a significant impact on mortality after adjustment. Patients with cirrhosis had higher mortality risks, while those undergoing chemotherapy had reduced mortality (AHRs of 0.92 for overall and 0.94 for cancer-related mortality). Surgery was associated with a substantial reduction in mortality, with AHRs of 0.60 for both overall and cancer-related mortality. Radiation therapy did not significantly affect mortality in this adjusted analysis.

The multivariate Cox proportional hazard regression analysis examining the interaction between age and fibrosis score (Table 4) on all-cause and cancer-specific mortality among US patients with hepatocellular carcinoma (HCC) from 2010 to 2021 highlights several key findings. For patients aged 00–39 with cirrhosis, the adjusted proportional hazard ratios (AHRs) for overall mortality (1.91) and cancer-specific mortality (1.55) were higher compared to those with mild/moderate fibrosis, although not statistically significant. Interestingly, for patients aged 40–59 and 60–79 with cirrhosis, the mortality risks were lower compared to the reference group, indicating no significant increase in hazard ratios. Notably, for patients aged 80+ with cirrhosis, there was a statistically significant reduction in overall mortality risk (AHR 0.57), though the reduction in cancer-specific mortality was not significant (AHR 0.71). These results suggest that while cirrhosis generally increases mortality risk in younger patients, the effect diminishes with age, potentially due to other age-related factors influencing survival.

## 4. Discussion

The analysis of US patients diagnosed with HCC between 2010 and 2021 reveals significant demographic and clinicopathologic characteristics. Out of 15,796 patients, 77.23% were male, with most diagnoses occurring between ages 60–79 (59.62%) and 40–59 (33.90%). Half of the patients were married (50.31%), and the majority were diagnosed at a localized tumor stage (61.02%). Non-Hispanic Whites were the largest racial group (47.78%), followed by Hispanics (23.31%). Most patients lived in metropolitan areas with populations of 1 million or more (62.95%). Income data showed many patients had annual incomes between $60,000 and $99,999, and 81.74% had cirrhosis. Surgery was the predominant treatment (94.44%), with chemotherapy (44.25%) and radiation therapy (12.42%) also utilized. Diagnosis numbers fluctuated, notably declining in 2020 and 2021. Males, older age, unmarried status, and advanced tumor stages significantly increase mortality. Chemotherapy and surgery were associated with reduced mortality. Interaction analysis showed reduced risks for those aged 80+, suggesting that the impact of cirrhosis diminishes with age.

The observed male predominance (77.23%) is consistent with global data indicating a higher incidence of HCC in males, often attributed to higher rates of chronic hepatitis B and C infections, alcohol consumption, and metabolic syndromes among men. Similarly, the majority of patients being diagnosed between the ages of 60 and 79 (59.62%) match the established understanding that HCC typically presents in older adults, reflecting the prolonged latency period required for chronic liver disease to progress to cancer [12,13]. The distribution of tumor stage at diagnosis, with 61.02% of cases being localized, is supported by other studies indicating that advancements in imaging techniques and surveillance programs have improved early detection rates of HCC. The racial distribution in this cohort, with Non-Hispanic Whites comprising the largest group (47.78%), followed by Hispanics (23.31%), is in line with national cancer statistics that highlight a higher prevalence of HCC among these populations, potentially due to variations in risk factor exposures and healthcare access [14,15].

Geographical living area data showing that a significant proportion of patients resided in metropolitan areas with a population of 1 million or more (62.95%) may reflect better access to healthcare facilities and diagnostic services in urban settings. Income data indicating substantial numbers of patients with annual incomes between $60,000 and $99,999 correspond with findings from socioeconomic studies that link higher income levels with increased access to healthcare and preventive services, potentially leading to earlier detection and treatment of HCC [16]. The high percentage of patients with cirrhosis (81.74%) corroborates the well-established association between cirrhosis and HCC development, as cirrhosis is a known major risk factor for the disease [17].

The predominant use of surgery (94.44%) as a treatment method aligns with clinical guidelines recommending surgical resection or liver transplantation as primary curative approaches for eligible HCC patients. The utilization of chemotherapy (44.25%) and radiation therapy (12.42%) is also consistent with current therapeutic strategies, where these modalities are often employed in cases where surgery is not feasible or as adjuvant treatments [17,18]. The fluctuation in yearly diagnosis numbers, with a noticeable decline in 2020 and 2021, can be attributed to the COVID-19 pandemic, which disrupted healthcare services and led to delays in cancer diagnoses and treatments. This observation is supported by numerous reports highlighting the impact of the pandemic on routine cancer care and diagnostic procedures [19,20,21].

We found that males had a higher mortality compared to females, which is consistent with other studies indicating a worse prognosis for men, and this could be attributed to a higher prevalence of risk factors such as hepatitis infections, alcohol use, and smoking in males [22,23]. The association of older age at diagnosis with increased mortality, particularly for those aged 80+, corroborates the established understanding that age is a critical determinant of HCC prognosis due to the cumulative effect of comorbidities and reduced physiological resilience in older patients [24]. The elevated mortality risks among unmarried patients, including single, divorced/separated, and widowed individuals, compared to married patients, reflect findings from previous studies that highlight the beneficial impact of marital support on cancer outcomes. This support can facilitate better adherence to treatment and follow-up care, contributing to improved survival rates [25,26].

Advanced tumor stages significantly elevate mortality risks, which is well-documented in the literature. Early detection and treatment are crucial for improving survival rates in HCC, as advanced stages are often less responsive to treatment and associated with poor prognosis [27]. The higher mortality risks associated with cirrhosis align with the well-known link between cirrhosis and HCC progression and complications [28]. The reduced mortality in patients undergoing chemotherapy (AHRs of 0.92 for overall and 0.94 for cancer-related mortality) and the substantial reduction in those undergoing surgery (AHRs of 0.60 for both) are consistent with clinical guidelines that emphasize the importance of aggressive treatment in suitable candidates to improve survival outcomes [17]. The finding that radiation therapy did not significantly affect mortality in this adjusted analysis may reflect the limited role of radiation in the management of HCC, which is typically reserved for specific cases where other treatments are not feasible [29,30].

For patients aged 80+ with cirrhosis, the statistically significant reduction in overall mortality risk is noteworthy. This finding suggests that other age-related factors, such as increased healthcare utilization and comorbidity management in older adults, might mitigate the adverse effects of cirrhosis on mortality. Studies have shown that elderly patients often receive more comprehensive care, which can contribute to improved outcomes despite the presence of severe fibrosis [31,32]. The reduction in cancer-specific mortality for patients aged 80+ with cirrhosis, although not statistically significant, supports the notion that the impact of cirrhosis on cancer progression might be less pronounced in very elderly patients, possibly due to competing risks from other age-related conditions that overshadow the cancer prognosis. This aligns with the literature indicating that the relative impact of HCC on mortality decreases with age, as older patients are more likely to die from non-cancer-related causes [33]. These results highlight the nuanced relationship between age, fibrosis, and HCC mortality. While cirrhosis did not increase mortality risk in younger patients, its effect diminishes with age, suggesting that personalized treatment approaches considering both age and fibrosis severity are essential for optimizing patient outcomes.

This study has several limitations inherent to the use of SEER data. Important clinical variables such as liver function status (e.g., Child–Pugh or MELD score), viral hepatitis status, alcohol use, BMI, and detailed comorbidity data are not available in the database. These unmeasured confounders could influence both survival outcomes and fibrosis progression, potentially biasing the observed associations. Although fibrosis staging data in SEER are limited and only available for a subset of patients, our analysis utilized the available fibrosis score variable as a surrogate indicator for cirrhosis. The absence of standardized measures of liver function or histological grading is a recognized limitation inherent to large population-based databases like SEER. Additionally, SEER does not capture detailed etiologic information regarding liver disease, such as hepatitis B or C infection, alcohol-related liver disease, or non-alcoholic steatohepatitis (NASH), limiting our ability to stratify patients by viral versus non-viral causes of HCC.

While the majority of patients were aged between 40 and 79 years (93.52%), only a small proportion (5.26%) were over 80 years old. The percentage of patients over 80 was not explicitly reported, which limits detailed interpretation for this subgroup. Given this small representation, the potential opposing effects of advanced age versus liver fibrosis on mortality should be interpreted with caution, as the influence of very old age in the analysis is limited and may not substantively affect the overall conclusions.

Overall, these findings suggest that clinical decision-making for HCC should consider not only the presence of cirrhosis but also the patient’s age. For younger patients (<60), cirrhosis remains a strong adverse prognostic factor and may warrant more aggressive surveillance and curative intent interventions. However, in patients aged 80 and above, the attenuated impact of cirrhosis on mortality suggests that other age-related risks may dominate the clinical picture. This raises the need for individualized, age-tailored treatment algorithms that balance oncologic benefits with life expectancy and comorbidities.

## 5. Conclusions

In this US population-based study on HCC mortality, significant male predominance and higher diagnoses among older adults align with global trends. Marriage appears to be protective, while advanced tumor stages and cirrhosis significantly increase mortality risks. The reduction in mortality with surgery and chemotherapy underscores their critical roles in HCC management. Notably, the interaction analysis between age and fibrosis score indicates that its impact diminishes with age, particularly among those aged 80 and above. These findings emphasize the need for personalized treatment strategies that account for both age and fibrosis severity to optimize outcomes for HCC patients.

## Figures and Tables

**Table 1 diseases-13-00256-t001:** Demographic and clinicopathologic characteristics of US patients diagnosed with hepatocellular carcinoma between 2010 and 2021.

Characteristics		
Total	*N*	%
	15,796	100
Gender		
Female	3597	22.77
Male	12,199	77.23
Age at Diagnosis, years		
00–39	193	1.22
40–59	5355	33.90
60–79	9417	59.62
80+	831	5.26
Marital Status		
Married	7947	50.31
Single	4222	26.73
Divorced/separated	2425	15.35
Widowed	1202	7.61
Tumor Stage		
Localized	9638	61.02
Regional	4572	28.94
Distant	1586	10.04
Race		
Non-Hispanic white	7548	47.78
Non-Hispanic black	1736	10.99
Hispanic	3682	23.31
Other	2830	17.92
Living Area		
Counties in metropolitan areas of 1 million persons	9944	62.95
Counties in metropolitan areas of 250,000 to 1 million persons	3488	22.08
Counties in metropolitan areas of 250,000 persons	1053	6.67
Nonmetropolitan counties adjacent to a metropolitan area	784	4.96
Nonmetropolitan counties not adjacent to a metropolitan area	527	3.34
Income Per Year		
<$40,000	139	0.88
$40,000–59,999	2151	13.62
$60,000–79,999	5648	35.76
$80,000–99,999	4822	30.53
$100,000+	3036	19.22
Fibrosis Score		
Mild/moderate fibrosis	2844	18.26
Cirrhosis	12,912	81.74
Radiation		
No	13,834	87.58
Yes	1962	12.42
Chemotherapy		
No	8807	55.75
Yes	6989	44.25
Surgery		
No	878	5.56
Yes	14,918	94.44
Year of Diagnosis		
2010	1235	7.82
2011	1395	8.83
2012	1500	9.50
2013	1463	9.26
2014	1551	9.82
2015	1655	10.48
2016	1535	9.72
2017	1394	8.83
2018	1197	7.58
2019	1205	7.63
2020	831	5.26
2021	835	5.29

**Table 2 diseases-13-00256-t002:** Crude analysis of factors associated with all-cause mortality and cancer-related mortality among US patients diagnosed with hepatocellular carcinoma between 2010 and 2021.

Characteristics	Overall Mortality. Crude Proportional Hazard Ratio (95% Confidence Interval)	Cancer-Related Mortality. Crude Proportional Hazard Ratio (95% Confidence Interval)
Gender		
Female	1(reference)	1(reference)
Male	1.19(1.13–1.25) **	1.20(1.12–1.27) **
Age at diagnosis, years		
00–39	1(reference)	1(reference)
40–59	1.77(1.41–2.24) **	1.71(1.33–2.21) **
60–79	1.86(1.48–2.34) **	1.79(1.39–2.31) **
80+	3.08(2.41–3.93) **	2.78(2.12–3.64) **
Marital status		
Married	1(reference)	1(reference)
Single		1.25(1.19–1.32) **
Divorced/separated		1.31(1.23–1.39) **
Widowed		1.34(1.23–1.45) **
Tumor stage		
Localized	1(reference)	1(reference)
Regional	2.22(2.12–2.32) **	2.53(2.41–2.66) **
Distant	4.92(4.61–5.26) **	5.74(5.34–6.17) **
Race		
Non-Hispanic white	1(reference)	1(reference)
Non-Hispanic black	1.07(1.01–1.14) *	1.05(0.98–1.13)
Hispanic	0.99(0.94–1.04)	0.99(0.93–1.04)
Other	0.78(0.74–0.83) **	0.77(0.72–0.82) **
Living area		
Counties in metropolitan areas of 1 million persons	1(reference)	1(reference)
Counties in metropolitan areas of 250,000 to 1 million persons	1.12(1.06–1.17) **	1.15(1.08–1.21) **
Counties in metropolitan areas of 250,000 persons	1.20(1.03–1.21) **	1.17(1.07–1.28) **
Nonmetropolitan counties adjacent to a metropolitan area	1.21(1.11–1.33) **	1.27(1.15- 1.40) **
Nonmetropolitan counties not adjacent to a metropolitan area	1.08(0.96–1.21)	1.12(0.99–1.27)
Income per year		
**<**$40,000	1(reference)	1(reference)
$40,000–59,999	0.99(0.80–1.23)	1.03(0.81–1.30)
$60,000–79,999	1.12(1.03–1.21) **	0.89(0.70–1.12)
$80,000–99,999	1.21(1.11–1.33) **	0.83(0.66–1.06)
$100,000+	0.70(0.57–0.87) **	0.70(0.55–0.88) **
Fibrosis score		
Mild/moderate fibrosis	1(reference)	1(reference)
Cirrhosis	1.28(1.21–1.35) **	1.28(1.20–1.36) **
Radiation		
No	1(reference)	1(reference)
Yes	1.26(1.18–1.33) **	1.33(1.24–1.41) **
Chemotherapy		
No	1(reference)	1(reference)
Yes	1.06(1.02–1.10) **	1.11(1.06–1.16) **
Surgery		
No	1(reference)	1(reference)
Yes	0.48(0.44–0.52) **	0.47(0.43–0.51) **

* *p* < 0.05, ** *p* < 0.01.

**Table 3 diseases-13-00256-t003:** Multivariate Cox proportional hazard regression analyses of factors affecting all-cause mortality and cancer-related mortality among US patients diagnosed with hepatocellular carcinoma between 2010 and 2021.

Characteristics	Overall Mortality. Adjusted Proportional Hazard Ratio (95% Confidence Interval)	Cancer-Related Mortality. Adjusted Proportional Hazard Ratio (95% Confidence Interval)
Gender		
Female	1(reference)	1(reference)
Male	1.18(1.12–1.25) **	1.17(1.11–1.24) **
Age at diagnosis, years		
00–39	1(reference)	1(reference)
40–59	1.68(1.33–2.12) **	1.62(1.25–2.09) **
60–79	1.85(1.47–2.34) **	1.79(1.38–2.31) **
80+	3.25(2.54–4.16) **	2.95(2.25–3.88) **
Marital status		
Married	1(reference)	1(reference)
Single	1.26(1.20–1.33) **	1.22(1.16–1.29) **
Divorced/separated	1.26(1.19–1.34) **	1.23(1.15–1.31) **
Widowed	1.33(1.23–1.45) **	1.30(1.19–1.42) **
Tumor stage		
Localized	1(reference)	1(reference)
Regional	2.17(2.08–2.27) **	2.46(2.34–2.58) **
Distant	4.81(4.49–5.15) **	5.57(5.18–6.00) **
Race		
Non-Hispanic white	1(reference)	1(reference)
Non-Hispanic black	1.02(0.96–1.10)	1.02(0.94–1.10)
Hispanic	1.04(0.99–1.10)	1.05(0.99–1.11)
Other	0.93(0.87–0.98) *	0.93(0.87–0.99) *
Living area		
Counties in metropolitan areas of 1 million persons	1(reference)	1(reference)
Counties in metropolitan areas of 250,000 to 1 million persons	1.05(0.99–1.11)	1.08(1.01–1.14) *
Counties in metropolitan areas of 250,000 persons	0.99(0.91–1.09)	1.03(0.94–1.14)
Nonmetropolitan counties adjacent to a metropolitan area	1.09(0.98–1.21)	1.14(1.02–1.28) *
Nonmetropolitan counties not adjacent to a metropolitan area	0.98(0.87–1.12)	1.02(0.89–1.18)
Income per year		
<$40,000	1(reference)	1(reference)
$40,000–59,999	1.04(0.84–1.29)	1.08(0.85–1.38)
$60,000–79,999	0.95(0.76–1.19)	1.01(0.79–1.29)
$80,000–99,999	0.92(0.73–1.14)	0.94(0.73–1.21)
$100,000+	0.84(0.67–1.05)	0.87(0.68–1.13)
Fibrosis score		
Mild/moderate fibrosis	1(reference)	1(reference)
Cirrhosis	1.26(1.20–1.34) **	1.25(1.18–1.34) **
Radiation		
No	1(reference)	1(reference)
Yes	1.01(0.95–1.07)	1.05(0.98–1.12)
Chemotherapy		
No	1(reference)	1(reference)
Yes	0.92(0.88–0.96) **	0.94(0.90–0.99) *
Surgery		
No	1(reference)	1(reference)
Yes	0.60(0.56–0.65) **	0.60(0.55–0.66) **

* *p* < 0.05, ** *p* < 0.01.

**Table 4 diseases-13-00256-t004:** Multivariate Cox proportional hazard regression analyses of factors affecting all-cause mortality in hepatocellular carcinoma-related mortality among US patients between 2010 and 2021, taking into account the interaction between age and fibrosis score.

Fibrosis Score and Age (Fibrosis Score#Age)	Overall Mortality	Cancer-Specific Mortality
Mild/moderate fibrosis # 00–39	1(reference)	1(reference)
Cirrhosis # 00–39	1.91(0.19–3.05)	1.55(0.93–2.57)
Cirrhosis # 40–59	0.69(0.42–1.11)	0.80(0.48–1.35)
Cirrhosis # 60–79	0.66(0.41–1.06)	0.83(0.49–1.38)
Cirrhosis 80+	0.57(0.35–0.95) *	0.71(0.41–1.23)

* *p* < 0.05.

## Data Availability

The data supporting the findings of this study are publicly available from the SEER Program at https://seer.cancer.gov (accessed on 15 June 2025).

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
