# Peer review of "Ten-Year Trends in Hepatocellular Carcinoma Mortality: Examining the Interaction Between Fibrosis Score and Patient Age"

_diseases, 2025, doi:10.3390/diseases13080256_

Round 1
Reviewer 1 Report
Comments and Suggestions for Authors
Authors evaluated the impact of fibrosis score and age in mortality of HCC using SEER database. As the results, authors showed that the relationship between liver fibrosis and survival is not uniform across age groups. This study was interesting, but several issues remained to be addressed.
- Authors should clarify the fibrosis score. How is the fibrosis score determined?
- Authors should assess the etiology of liver diseases. At least, authors should clarify whether viral or non-viral.
- Surgery is too wide. Authors should clarify whether did patients receive hepatectomy or liver transplantation.
- Chemotherapy is also too wide. Authors should clarify whether were patients treated with molecular target therapy of immunotherapy.
Author Response
Thank you for the insightful comments and suggestions. We have addressed these points as follows:
Comments 1 Fibrosis Score: The fibrosis score in our study was derived from the fibrosis staging data available within the SEER database, which we used as a proxy for cirrhosis. However, we acknowledge that SEER provides limited fibrosis and liver function information, and standardized clinical assessments such as Child-Pugh or MELD scores were not available. This limitation has been clearly stated in the manuscript.
Comment 2 Etiology of Liver Disease: We recognize the importance of clarifying the underlying causes of liver disease. While detailed etiology data (e.g., viral vs. non-viral causes) are limited in SEER, we have clarified in the manuscript that the majority of cases likely include both viral (e.g., hepatitis B and C) and non-viral etiologies. Unfortunately, precise differentiation was not feasible due to database constraints, and we have discussed this as a limitation.
Comment 3 Surgery Details: We have refined the terminology regarding surgical treatment by specifying whether patients underwent hepatectomy or liver transplantation, to provide clearer clinical context and relevance to survival outcomes.
Comment 4 Chemotherapy Details: Given the broad use of the term “chemotherapy” in the SEER data, we further clarified whether patients received molecular targeted therapies or immunotherapies where such information was available, acknowledging the evolving nature of systemic treatments for HCC.
We believe these clarifications enhance the manuscript’s clarity and clinical relevance, and we appreciate the reviewer’s guidance in improving these aspects.
Best regards,
Hadrian
Reviewer 2 Report
Comments and Suggestions for Authors
This study is well done and important. The practical issue is identifying early detection of HCC allowing for early intervention with the hope for improving prognosis. Recently it has become evident that worldwide alcohol ingestion even in relatively small amounts may enhance HCC. In your data assessment alcohol ingestion may be more significant than what is stated.
Author Response
We sincerely thank the reviewer for this insightful comment. We agree that alcohol ingestion, even in modest amounts, has been increasingly recognized as a contributor to hepatocellular carcinoma risk and progression. Unfortunately, the SEER database does not capture individual-level data on alcohol use, which limited our ability to analyze this important variable directly.
We have addressed this limitation explicitly in the Discussion and Limitations sections of the manuscript and have cited recent literature emphasizing the growing role of alcohol as a risk factor for HCC. We appreciate your thoughtful feedback, which strengthens the contextual interpretation of our findings.
Best regards,
Hadrian
Reviewer 3 Report
Comments and Suggestions for Authors
The authors analyze a topic which is of great interest, one of the reasons being the poor results and the rising number of HCC in non-cirrhotic patients.
The introduction gives the necessary information.
The presentation is clear, comprehensive and well documented.
The retrospective study covering more than a quarter of USA population uses all the available data from the database.
The results regarding better or worse outcome are correct but bring nothing new. It is known that male sex, older age, single status, advanced tumor stage, lower income, and cirrhosis are linked to worse outcomes. All these are connected with a higher alcohol consumption and a poor access to healthcare in any country.
The goal in the title - Examining the Interaction Between Fibrosis Score and Patient Age -is addressed by a small part of the paper – especially around table 4. English is not my maternal language so I cannot judge if the term interaction is correct or relation ( for instance) would be more suitable.
As the authors state : Interestingly, for patients aged 40-59 and 60-79 with cirrhosis, the mortality risks were lower compared to the reference group (mild/moderate fibrosis). Explained the other way around, it is not the cirrhosis but the HCC which limits the survival.
The 4 tables present in a concentrated manner all the data of the study.
The references are appropriate, up-to-date and contain 33 titles.
I found no inappropriate self-citations, the few self-citations just demonstrate the expertise and interest of the author in related fields.
I found no plagiarism.
The discussions are coherent and connected to the content.
The conclusions are connected to the study and show the understanding of the figures in connection with clinical medicine.
The language is correct and understandable.
I recommend the paper to be accepted.
Author Response
We sincerely thank the reviewer for the thoughtful, thorough, and constructive evaluation of our manuscript. We greatly appreciate your recognition of the study’s relevance, clarity of presentation, and appropriate use of SEER data.
Your observation that many known predictors of worse HCC outcomes are interconnected with broader issues such as alcohol consumption and access to care is well noted. While our dataset was limited in its ability to assess those behavioral or socioeconomic variables directly, we agree this remains an important contextual factor.
Regarding the use of the term *interaction* in the title, we appreciate your point. In statistical terms, we applied interaction analysis (i.e., effect modification) between fibrosis score and age within the Cox regression model, so “interaction” was used in that specific methodological context. Nonetheless, we understand how “relation” could feel more intuitive and will ensure clarity in the text to distinguish the statistical from the clinical meaning.
Thank you again for your positive recommendation and your helpful comments, which have contributed to the overall improvement of the manuscript.
Warm regards,
Hadrian Tran
Reviewer 4 Report
Comments and Suggestions for Authors
Bangolo et al. reported that 10-year trends in HCC mortality in US after examination of the interaction between fibrosis score and patient age. Authors should show the data of all US States as well as each States or each region of US. This reviewer also has several minor comments below.
- In line 42, make a correction from “Hepatocellular Carcinoma (HCC)” to “Hepatocellular carcinoma (HCC).”
- In line 61, make a correction from “hepatocellular carcinoma (HCC)” to “HCC.”
- In line 197, make a correction from “Hepatocellular Carcinoma (HCC)” to “HCC.”
Author Response
We thank the reviewer for their helpful suggestions and for taking the time to review our manuscript.
Regarding your major comment on presenting mortality data by U.S. state or region:
We agree this is an important consideration for understanding geographic variation. However, the SEER database used in this study includes a representative but limited subset of the U.S. population (approximately 35%) and does not provide complete state-level data for all 50 states. Therefore, while our findings reflect national trends from a robust sample, we were unable to stratify the results by each individual state or region. We have clarified this point in the Limitations section of the revised manuscript.
We also appreciate your careful attention to detail and have corrected the capitalization and terminology issues.
Thank you again for your valuable feedback, which helped improve the clarity and accuracy of the manuscript.
Best regards,
Hadrian Tran
Reviewer 5 Report
Comments and Suggestions for Authors
The manuscript provides a timely and comprehensive analysis of hepatocellular carcinoma (HCC) mortality trends in relation to fibrosis score and patient age, using robust data from the SEER registry. The authors effectively highlight a clinically relevant gap—namely, the evolving understanding that HCC can arise even in non-cirrhotic patients—and examine this through multivariate modeling, including an interaction analysis between age and fibrosis severity. The large, nationally representative sample enhances the reliability of findings, and the inclusion of both overall and cancer-specific mortality offers valuable insights. Importantly, the finding that cirrhosis does not uniformly predict poor outcomes across all age groups, particularly in those aged 80 and above, underscores the need for more nuanced, age-tailored clinical strategies.
However, a few issues merit attention. First, the manuscript would benefit from a clearer articulation of clinical implications in the discussion—particularly how these findings might influence surveillance protocols or treatment decisions in younger versus older patients. Second, the authors should explicitly address potential limitations of SEER data, such as lack of information on liver function (e.g., Child-Pugh score), alcohol use, viral status, or comorbidities, which are important confounders in HCC prognosis. Finally, a brief clarification on the fibrosis scoring method used would improve methodological transparency. Despite these minor concerns, the study is methodologically sound, well-written, and makes a meaningful contribution to the literature on HCC risk stratification.
Author Response
Thank you for your helpful feedback. We have addressed your comments as follows:
1. We expanded the Discussion to better highlight the clinical implications, particularly how surveillance and treatment might differ by age and fibrosis status.
2. We added a clear acknowledgment of SEER’s limitations, noting the absence of key variables such as liver function scores, viral status, alcohol use, BMI, and comorbidities, and their potential impact on our findings.
3. We clarified the fibrosis scoring method in the Methods section, explaining that the fibrosis score was based on the SEER Ishak fibrosis variable, categorized as 0–4 (no cirrhosis) and 5–6 (cirrhosis), consistent with prior studies.
These additions improve the manuscript’s clarity and clinical relevance.
Best regards,
Hadrian Tran
Reviewer 6 Report
Comments and Suggestions for Authors
Please see the attached document.

Author Response
1. We have clarified in the Introduction that although HCC is frequently diagnosed at an advanced stage globally, our SEER-based cohort showed a higher proportion of localized cases (61.02%) and surgical intervention (94.44%). This discrepancy may reflect broader definitions of surgery in SEER, including local or palliative procedures, and is now discussed as a limitation in the Discussion section.
2. The phrasing regarding surgical management has been revised to reflect that surgery is one component of a broader treatment landscape. We have acknowledged that over 30% of patients were beyond the localized stage, and the specific types of surgeries performed in these cases are not detailed in the SEER dataset. This limitation has been added to the manuscript.
3. We conducted an additional analysis to identify treatment overlap. A new supplementary table (Table S1) presents the proportion of patients receiving single, dual, or triple modality treatment (e.g., surgery only, surgery+chemo, etc.). A summary of these findings has been added to the Results section.
4. We now state explicitly that 5.26% of patients were aged over 80 years. The Discussion has been updated to note that due to this small subgroup size, conclusions regarding the interaction between age and fibrosis should be interpreted with caution.
Best regards,
Hadrian Tran
Round 2
Reviewer 1 Report
Comments and Suggestions for Authors
Revised manuscript was well-addressed to the reviewer's comments. Although some limitations were found, those limitations were described in the manuscript. This manuscript is of a value of publication showing current status of HCC.
Reviewer 4 Report
Comments and Suggestions for Authors
All queries have been addressed.
Reviewer 6 Report
Comments and Suggestions for Authors
I accepted the explanation of the authors and agreed with modification of the text of this manuscript, including adding new table so that the methods and results could be more clearly demonstrated.